# Adipokines as Immune Cell Modulators in Multiple Sclerosis

**DOI:** 10.3390/ijms221910845

**Published:** 2021-10-07

**Authors:** Merel Rijnsburger, Niek Djuric, Inge A. Mulder, Helga E. de Vries

**Affiliations:** 1Department of Molecular Cell Biology and Immunology, MS Center Amsterdam, Amsterdam Neuroscience, Amsterdam UMC, Vrije Universiteit Amsterdam, de Boelelaan 1108, 1181 HZ Amsterdam, The Netherlands; n.djuric@gmail.com (N.D.); he.devries@amsterdamumc.nl (H.E.d.V.); 2Department of Biomedical Engineering and Physics, Amsterdam UMC, University of Amsterdam, Meibergdreef 9, 1105 AZ Amsterdam, The Netherlands; i.mulder1@amsterdamumc.nl

**Keywords:** obesity, multiple sclerosis, adipokines, neuroinflammation, demyelination, neurodegeneration, immunology

## Abstract

Multiple sclerosis (MS), a chronic inflammatory and demyelinating disease of the central nervous system (CNS), is a major clinical and societal problem, which has a tremendous impact on the life of patients and their proxies. Current immunomodulatory and anti-inflammatory therapies prove to be relatively effective; however, they fail to concomitantly stop ongoing neurological deterioration and do not reverse acquired disability. The proportion to which genetic and environmental factors contribute to the etiology of MS is still incompletely understood; however, a recent association between MS etiology and obesity was shown, with obesity greatly increasing the risk of developing MS. An altered balance of adipokines, which are white adipose tissue (WAT) hormones, plays an important role in the low-grade chronic inflammation during obesity by their pervasive modification of local and systemic inflammation. Vice versa, inflammatory factors secreted by immune cells affect adipokine function. To explore the role of adipokines in MS pathology, we will here review the reciprocal effects of adipokines and immune cells and summarize alterations in adipokine levels in MS patient cohorts. Finally, we will discuss proof-of-concept studies demonstrating the therapeutic potential of adipokines to target both neuroinflammation and neurodegeneration processes in MS.

## 1. Introduction

Multiple sclerosis (MS) is one of the most common causes of permanent disability in young adults in Western countries and affects about 2.8 million patients worldwide [1]. MS is considered a chronic inflammatory and demyelinating disease of the central nervous system (CNS), resulting in severe physical problems and cognitive decline. To date, no strategies exist that fight the chronic neuroinflammation in MS and concomitantly promote neuroprotection.

Various genetic, environmental, and lifestyle factors are believed to contribute to disease onset and progression. Accumulating evidence suggests that metabolic changes such as increased body mass index (BMI (kg/m^2^) and obesity play a major role in the risk of developing MS [2,3,4,5,6], and obesity has been linked to higher disability, increased (neuro)inflammation, and even reduced gray matter volume [7,8]. The activation and chronicity of the involved pro-inflammatory cascade in obesity is driven by a disturbed balance of pro-inflammatory and anti-inflammatory hormones from white adipose tissue (WAT)—so-called adipokines—into the circulation. Primarily described as metabolic modulators, these adipokines also modify local and systemic inflammation and vice versa; inflammatory factors disrupt adipokine signal transduction and thereby impair their function [9,10,11,12]. Thus, changes in adipokine levels and signaling thereof may affect the onset of disease due to (genetic) overweight/obesity, and they may also trigger disease processes. Alternatively, alterations in adipokines could be secondary to changes in immune responses during MS, together creating a detrimental feedforward loop between pro-inflammatory adipokines and cytokines, resulting in chronic inflammation and eventually neurodegeneration (Figure 1).

We will here provide a comprehensive overview of recent findings on adipokine alterations in MS patients and their involvement in pathological processes evidenced by in vitro as well as animal studies. We further provide new insights on how to go forward to use adipokines as disease biomarkers and as a novel potential therapeutic strategy, fighting both neuroinflammation and neurodegeneration.

## 2. MS Pathophysiology, Therapies, and Risk Factors

### 2.1. Pathophysiology

The classic clinical presentation of MS includes blurred vision (optic neuritis), impaired sensation, weakness, and/or ataxia (partial myelitis), focal sensory disturbance and brainstem syndromes (vertigo, hearing loss, facial sensory disturbance). In most MS patients, the disease starts with a relapsing–remitting course (RRMS), which presents as clinical relapses with near or complete recovery. Over time, the recovery after relapses may become incomplete, and disability often accumulates. Approximately 20% of patients with RRMS develop progressive neurologic decline later in the disease and transition to secondary progressive MS (SPMS). A minority of the patients (10–15%) suffer from primary progressive MS (PPMS) and experience uninterrupted progression from disease onset [13].

Despite the clinical distinction between disease types, neuroinflammation is a key component of all disease phenotypes. The characteristic pathological hallmark of MS is perivascular inflammatory lesions, which, in white matter lesions, is the result of complex pathological processes including dysfunction and inflammation of the blood–brain barrier (BBB) and immune cell migration. This chronic neuroinflammatory status subsequently induces demyelination, axonal damage, and neurodegeneration [13]. Although MS has been classically thought of as a typical white matter disorder, the involvement of gray matter regions in the demyelinating process was acknowledged in early pathology studies. The pathology of gray matter lesions differs from that of white matter lesions in that significant lymphocyte infiltration and BBB disruption have so far not been detected in these lesions. However, a number of studies now indicate that chronic, compartmentalized inflammation of the nearby meninges is likely to drive many aspects of cortical pathology. More in-depth elaboration on the different inflammatory pathways activated in MS can be found elsewhere [14,15,16,17,18].

### 2.2. Disease-Modifying Treatments

In today’s clinic, multiple disease-modifying treatments (DMTs) are available for the treatment of RRMS, targeting various disease aspects and with a large range of efficacy [19]. The first DMT, IFN-β, was approved in 1993 and is still widely prescribed [20]. Since then, 10 classes of DMTs are FDA (Food and Drug Administration) approved for MS and comprise interferons, sphingosine 1-phosphate (S1P) receptor modulators, cladribine, teriflunomide, mitoxantrone, glatiramer acetate, fumarates, CD20 antibodies, natalizumab, and alemtuzumab. These DMTs have a wide range of mechanism of actions, administration routes, and efficacy, but all are immunomodulatory. The treatment effect of IFN-β is related to several overlapping mechanisms including the downregulation of major histocompatibility complex (MHC) class II expression present on antigen-presenting cells, production of interleukin 10 (IL-10), and inhibition of T cell migration into the brain [20]. S1P receptor modulators bind various subtypes of S1P receptors, which results in their internalization and sequestration of lymphocytes in lymph nodes. Cladribine, teriflunomide, and mitoxantrone interfere with DNA or protein synthesis, which leads to cell death. The glatiramer acetate structure resembles myelin basic protein and thus is hypothesized to act as a decoy, diverting an autoimmune response against myelin. Additionally, it has been shown to cause Th1/Th2 shift [21]. Fumarates such as dimethyl fumarate are believed to act both centrally by enhancing the nuclear factor erythroid 2 related factor 2 (Nrf2) transcriptional pathway, to counteract oxidative stress, and peripherally by shifting the T cells balance from Th1/Th17 phenotype to anti-inflammatory Th2 phenotype [22]. CD20 monoclonal antibodies selectively bind B cells that express the CD20 antigen, leading to cell destruction by complement dependent cytotoxicity as well as antibody-dependent, cell-mediated cytotoxicity [23,24,25]. Natalizumab is a monoclonal antibody that binds the α4 integrin subunit expressed on the surface of leukocytes and prevents their entry into the CNS. Alemtuzumab is a monoclonal antibody that binds CD52 on lymphocytes, leading to lymphocyte depletion [19]. The only FDA-approved DMT for treatment of PPMS is ocrelizumab. Ocrelizumab is hypothesized to treat progressive disease by reducing B cell-mediated inflammation that may lead to neurodegeneration. The recent discovery of meningeal B cell follicles in SPMS suggests that B cells may play a more significant role in progressive disease, potentially explaining the benefit seen with B cell-depleting treatments. However, this new strategy only shows satisfactory effects in patients with signs of inflammatory disease activity, which is less pronounced in PPMS [26]. Despite the proven reduction in relapse rate and disability of approved DMTs, treatments that effectively modulate inflammation without imposing an increased risk of infections, secondary autoimmunity, or fatal side effects, as well as treatments that improve established disability, are still needed. In addition, current treatments fail to suppress ongoing neuroinflammation and concomitantly halt neurodegeneration. In addition, the worldwide availability of effective drugs is limited due to high costs [27].

### 2.3. Obesity as Risk Factor for MS

The exact cause of MS remains unknown, but various genetic, lifestyle, and environmental factors are thought to contribute to disease onset. Amongst others, these include low sunlight exposure, childhood migration to high-risk countries, tobacco exposure, and viruses (such as the Epstein–Barr virus and human herpesvirus) [4].

Interestingly, increasing evidence suggests that metabolic changes such as increased BMI (kg/m^2^) and obesity, as a result of an altered lifestyle, also play a major role in MS development and progression. Obesity (BMI > 30) during adolescence, but not during childhood or adult life, is strongly associated with a higher risk of MS development. With a risk increase up to 96%, obesity outnumbers established heritable (risk increase of ≈40% [28]) and environmental risk factors [2,3,5,6]. Increased BMI is linked to higher disability in MS [29] and associated with reduced treatment response to IFN-β [30]. In RRMS patients with obesity, increased CSF levels of pro-inflammatory interleukin 6 (IL-6) and decreased levels of anti-inflammatory IL13 are reported [7]. Furthermore, each 1 kg/m^2^ higher BMI is shown to be correlated with reduced gray matter and brain parenchyma volume, and MS patients with BMI > 40 (morbidly obese) have a nearly 30 mL reduction in gray matter volume [8]. The detailed mechanistic pathways through which obesity is involved in MS pathogenesis are far from clear; however, these data strongly suggest that metabolic and immunological alterations that accompany increased body weight can promote disease pathogenesis and might even directly affect the brain.

From a pathophysiological point of view, MS and obesity have at least three distinctive pathways in common that are not mutually exclusive but may overlap: (1) chronic inflammation (i.e., increased circulating levels of TNF-α, IL-6, IL-1β, IFN-γ, and many others [31,32]), (2) endocrine alterations (disturbed secretion of adipokines), and (3) alterations in the gut microbiome (extensively reviewed elsewhere [33]). Here, we will review how chronic inflammation in obesity is driven by altered release of white adipose tissue hormones, the adipokines.

## 3. Adipokines, Inflammation, and MS

Overweight and obesity are associated with low-grade chronic inflammation of WAT, which is accompanied by an altered secretion of adipokines. These adipokines are WAT hormones and cytokines that have autocrine and paracrine functions and regulate a plethora of metabolic processes. Leptin was the first adipokine to be discovered in 1994, and since then, over 600 adipokines have been described [34]. Below, the function and implications of adipokines in MS are described (Section 3.7), which are primarily (although not exclusively) produced by the WAT and known to significantly affect the immune system. By far, the most studied ‘true adipokines’ are adiponectin, leptin, and resistin. More recently discovered adipocyte-derived hormones that are emerging as important players in immune regulation are chemerin, visfatin, and apelin [35,36]. Despite the strong evidence that adipokines play pivotal roles in immunity, the exact involvement of adipokines in MS pathophysiology remains largely unknown. Studies have been conducted in the past decades to establish a possible correlation with MS pathology, and alterations of both systemic as well as CSF adipokine levels have been reported, with different outcomes (Section 3.7), which will be reviewed below.

### 3.1. Adiponectin

Adiponectin, the most abundant adipokine in the human circulation, is produced as a monomer that circulates in serum as different oligomers. The oligomerization state and the tissue expression of adiponectin and its receptors (adiponectin receptor (AdipoR) 1 and 2, seven-transmembrane receptors) are linked to its biological activities, and thus, changes thereof could be of relevance for function [37]. Both receptors mediate adiponectin signaling via the activation of AMP-activated kinase (AMPK), which subsequently increases free fatty acid influx and β-oxidation, and peroxisome proliferator-activated receptor (PPAR)α signaling. Both pathways are involved in the major downstream anti-inflammatory effects of adiponectin in various target tissues [38]. Pro-inflammatory cytokines such as TNF-α, IL-1β, and IFN-γ dose- and time-dependently suppress adiponectin secretion by adipocytes, which probably accounts for the decreased plasma levels during chronic inflammation in obesity [39].

Adiponectin via its receptors exerts predominantly anti-inflammatory effects. For instance, adiponectin inhibits IFN-γ and IL-17 production in CD4^+^ T cells in obese mice by targeting intracellular metabolism [40]. In contrast, it is also shown that adiponectin induces a pro-inflammatory state in human macrophages, increasing the production of pro-inflammatory IL-6 and IFN-γ, as well as Th1 differentiation of CD4^+^ T Cells [41]. It is suggested that adiponectin induces a degree of inflammatory activation in macrophages that likely mediates tolerance to further pro-inflammatory stimuli. In endothelial cells, adiponectin limits TNF-α-induced expression of intracellular cell adhesion molecule 1 (ICAM-1) and consequently leukocyte rolling and adhesion [42]. Adiponectin’s protective effects on the endothelium are thought to be mediated via the induced expression of nitric oxide [37]. PPAR-γ and PPAR-α agonists, such as rosiglitazone, are recently discovered as anti-inflammatory agents. These compounds increase adiponectin expression and secretion by post-translational modification of HMW adiponectin as well as AdipoR expression in adipocytes, enhancing adiponectin’s action [43].

In MS, studies report lower levels in female RRMS patients in relapse phase versus controls [44], and higher levels during remission [45]. Furthermore, elevated adiponectin levels in early-onset female RRMS patients versus controls have been found [46]. Others observed decreased levels in MS versus controls [47,48], of which the latter showed it to be most apparent in SPMS. After correcting for gender, only a decrease was found in total MS female patients versus controls [48]. Adiponectin levels were shown to increase upon initiation of IFN-β therapy, but they were not associated with clinical or MRI disease activity or with treatment response, and thus, they are not useful as biomarker for disease activity [49]. Higher blood levels of total adiponectin in MS patients before the start of any treatment are reported and correlated with increased disability and progression [50]. In line, others found lower levels in patients with a milder disease course [51]. These results suggest that adiponectin represents a potential biomarker at the onset of disease to predict progression and severity. CSF levels of adiponectin can also be a good indicator for progression, since two studies measured an increase in CSF levels in MS versus controls. In CSF, higher adiponectin levels in PPMS compared to RRMS patients were found, and higher EDSS at baseline and a worse disease severity at 4.5-year follow-up were associated with elevated adiponectin levels. In addition, adiponectin levels correlated to IgG levels in the CSF, and the adiponectin oligomerization profile was altered in the CSF of MS patients, with a significant increase in the metabolically active HMW and MMW isoforms [52]. Although limited by sample size (four paired CSF and plasma samples), an increase in adipokine levels of patients compared to their asymptomatic co-twins is also described. However, due to a lack of correlation between CSF and plasma levels, they suggest a secondary, intrathecal synthesis of adiponectin [53].

### 3.2. Leptin

Leptin is an established pro-inflammatory hormone, and its increase during obesity is partly responsible for the associated low-grade inflammatory state. Leptin’s effects are widespread, targeting both innate and adaptive immune responses (recently reviewed elsewhere [54]). Six leptin receptor (LepRs) isoforms with different physiological roles have been identified [55]. Upon binding leptin, a cascade of signaling events is elicited, beginning with the activation of the constitutively receptor-associated Janus kinase-2 (JAK2), which is a tyrosine kinase. The activation of Jak2 stimulates the phosphorylation of multiple residues on the intracellular domain of LepR, which in turn can lead to activation of the extracellular signal-regulated kinase (ERK) cascade and phosphorylation of signal transducer and activator of transcription-3 (STAT3) and STAT5. The suppressor of cytokine signaling-3 (SOCS3) serves as a negative regulator of LepR signaling [56]. In human T cells, B cells, and monocytes, specific activation for each cell type caused a significant increase in LepR expression, and leptin treatment increases the cytokine production of IL-6, TNF-α, and IL-10 in activated B cells via the increased phosphorylation of STAT3 [57]. Furthermore, leptin stimulates myelin-specific T cell proliferation, reduces apoptosis, and promotes pro-inflammatory cytokine secretion. The proliferation of regulatory T (Treg) cells is also significantly inhibited after stimulation with leptin [58]. The effects of leptin on myelin-specific T cells are mediated by increased STAT3 and ERK1/2 phosphorylation. In contrast, the inhibiting effects on Treg cells are via the decreased phosphorylation of ERK1/2 [58].

In monocytes, leptin upregulates pro-inflammatory cytokines (IL-1, IL-6, IL-12, TNF-α, granulocyte colony-stimulating factor (G-CSF), and granulocyte-macrophage-CSF (GM-CSF)), stimulates the production of reactive oxygen species (ROS), and increases phagocytosis. Vice versa, pro-inflammatory cytokines (IL-1β, TNF-α, and IL-6) induce the secretion of leptin by mononuclear phagocytes [11,59]. Furthermore, rodent studies show that inflammatory stimuli such as lipopolysaccharide (LPS) dramatically increase circulating leptin levels [12]. In addition, in primary rat microglia, leptin treatment induces the expression of IL-1β and TNF-α and enhances LPS effects [60], and leptin led to large increases in IL-6 production in the BV2 microglial cell line via the upregulation of nuclear factor kappa-light-chain-enhancer of activated B cells (NF-κB) translocation [61], which is a key regulator of inflammatory immune responses and is involved in cytokine production. This reciprocal interaction between leptin and pro-inflammatory factors further drives the inflammatory response and could play a significant role in the pathology of MS.

In MS, leptin blood levels have been investigated by 16 studies [44,47,51,53,62,63,64,65,66,67,68,69,70,71,72,73]. In summary, out of these 16 studies, nine found significantly higher serum leptin levels in MS patients compared to healthy controls [44,47,51,62,64,66,67,71,72]. Six studies observed no difference [53,63,65,68,69,73], and one reported lower serum leptin levels [70]. Interestingly, eight out of the 10 studies did not detect differences during the remission phase of RRMS [53,63,65,67,68,69,73]. The remaining two studies reported higher levels in the MS groups [51,72]. In contrast, four out of seven studies observed significantly higher serum leptin levels in patients during relapse [44,47,66,71], two found lower levels [63,72], and one study no difference [67]. These data suggest that leptin levels might predict a relapse. In line, [73] showed that leptin levels indeed increased preceding a relapse. No clear association between serum leptin levels and EDSS scores was found in RRMS patients, as four out of the five studies reported no difference [49,69,73,74,75]. Of note, these four studies also reported no differences in leptin levels. In contrast, serum leptin and EDSS correlated positively in SPMS and PPMS patients [62,76]. Moreover, three studies showed a negative correlation between leptin and numbers of T-regulatory cells [47,64,66], and two studies showed positive correlations between leptin levels and pro-inflammatory cytokines (TNF-α, IL-1β, CRP, and IFN-γ) [64,66]. Two studies measured CSF leptin levels, of which one observed no differences (but also not in plasma) [53] and one study reported significantly higher leptin levels in RRMS patients in both serum and CSF [66].

The general trend in the data points toward increased leptin levels in MS patients and suggests that higher leptin levels associate with increased disease burden. Nevertheless, it should be noted that these studies were of varying sample size. Most study populations were matched/corrected for gender and BMI, and only 10 studies used patients that were not on disease-modifying treatments [44,51,64,66,67,68,69,71,72]. All of these parameters may influence the results. Nonetheless, the discussed clinical evidence illustrates that leptin plays a significant role in the pathophysiology of MS and indicates that possibilities for leptin as a treatment target should be further explored.

### 3.3. Resistin

Resistin (or ‘resistance to insulin’) was originally discovered in mice and named for its ability to resist (interfere with) insulin action. It belongs to a family of resistin-like molecules with distinct expression patterns and biological effects [77]. Circulating resistin has been positively correlated with C-reactive protein (CRP), TNF-α, and IL-6 in type 2 diabetes, rheumatoid arthritis, chronic kidney disease, sepsis, and coronary atherosclerosis [78]. The main source of mouse resistin is WAT adipocytes; however, human adipocytes produce little or no resistin; rather, it is produced by other cell populations, which include peripheral blood mononuclear cells (PBMCs), macrophages, and bone marrow cells [79]. The functional resistin receptor is adenylyl cyclase-associated protein 1 (CAP1) [80]. Additionally, resistin also competes with LPS for binding to Toll-like receptor 4 (TLR4), which is a key activator of inflammatory pathways [81].

Resistin is shown to increase the production of TNF-α, IL-6, IL-1β, and MCP-1 in human PBMCs and macrophages. Vice versa, when stimulated with pro-inflammatory cytokines, resistin levels in PBMCs increase, further enhancing its own activity in a positive feedback loop [82,83]. In human endothelial cells (ECs), the pro-inflammatory effects of resistin are antagonized by adiponectin, suppressing the expression of ICAM-1 and vascular cell adhesion molecule 1 (VCAM-1). Furthermore, resistin’s effects are shown to be mediated via nuclear translocation of NF-κB [82,83,84].

In MS, increased resistin levels are reported, as well as positive correlations with TNF-α and IL-1β [64] and EDSS [47,51], which would be in line with its described pro-inflammatory function. Others found no differences between patients with clinically isolated syndrome (CIS)—a first episode of neurologic symptoms that may or may not go on to develop in MS—and the different MS types [85]. Since resistin release is stimulated by pro-inflammatory cytokines and vice versa, and primarily produced by PBMCs in humans, increased resistin production by PBMCs of MS patients could be an important driver of chronic inflammation.

### 3.4. Chemerin

Chemerin is synthesized as a 163 amino acid precursor in the liver and WAT, and it is cleaved by an unknown protease to form pro-chemerin. Different isoforms of chemerin have distinct effects on immune cells, which complicates it to exactly pinpoint its role. Circulating chemerin levels are increased in various inflammatory diseases such as rheumatoid arthritis [86] and correlate to disease activity [87] and to several circulating pro-inflammatory markers, such as TNF-α, IL-6, and CRP [88,89]. Chemerins are ligands of chemokine-like receptor 1 (CMKLR1, also called Chemerin receptor 23, ChemR23), a G protein-coupled receptor expressed on various cell types. Chemerin is in direct competition with Resolvin E1 for CMKLR1 binding, which is described to promote the resolution of inflammation [90].

Chemerin promotes the initiation of inflammation by aggregating different immune cell types to inflammatory sites in the early stage of inflammation. At the resolution phase of inflammation, a protease released by macrophages and apoptotic cells transforms chemerin into an anti-inflammatory factor [91]. In vitro, it has been shown that pro-inflammatory (so-called M1) macrophages express mRNA for CMKLR1, while anti-inflammatory M2 macrophages do not. After CMKLR1 activation with Resolvin E1, M1 macrophages increase IL10 expression and adapted a resolution-like phenotype [92]. In contrast, active chemerin (i.e., chemerin 156) results in increased expression of pro-inflammatory cytokines (IL-1β, TNF-α, and IL-12) in J744A.1 macrophages co-cultured with tumor cells [93]. These mechanisms can provide a forward control loop through which the chemerin/CMKLR1 axis both self-activates and self-limits its action. On the contrary, chemerin does not have a direct effect on the strong upregulation of cytokine release of human and mice macrophages upon stimulation by LPS or LPS/IFN-γ [94].

Recently, two other chemerin receptors have been identified, G protein-coupled receptor-1 (GPR-1) and chemokine receptor-like 2 (CCRL2) [95,96], but more in-depth investigation is needed to clear current discrepancies and unravel chemerin’s effects on macrophages and other immune cells via these receptors.

Data on chemerin involvement in MS pathology are sparse; only two studies assessed chemerin levels in MS patients [97,98] but found no differences other than a positive association with BMI and increased levels in females, as described previously [99].

### 3.5. Visfatin

The adipokine visfatin (also known as extracellular nicotinamide phosphoribosyltransferase (NAMPT)) is highly expressed in visceral fat [100]. Visfatin is the rate-limiting enzyme for the synthesis of nicotinamide adenine dinucleotide (NAD) and thereby regulates intracellular metabolism. Extracellularly, it has been shown that visfatin binds the insulin receptor as well as activates TLR4. Studies have shown that the synthesis and secretion of visfatin is stimulated by IL-6 and TNF-α [101,102], whereas the anti-inflammatory PPAR-γ agonist rosiglitazone reduced visfatin expression by adipocytes [101].

Visfatin is thought to have various pro-inflammatory roles independent of its enzymatic activity, and it is required for lymphocyte development [103]. In both human and murine leukocytes in vitro, visfatin stimulates the production of TNF-α, IL-6, IL-1β, and IL-10 as well as chemotaxis via the MAPK/p38 signaling pathway [104,105]. In line, FK866, a specific, non-competitive visfatin inhibitor, dose-dependently inhibited LPS-induced IL-6, IL-1β, inducible nitric oxide synthase (iNOS), nitric oxide, and ROS levels in BV2 microglial cells, probably via the suppression of NF-κB phosphorylation [106]. In human EC, visfatin upregulates the gene expression of multiple chemokines and cytokines (C–C motif ligand CCL2, IL-6, GM-CSF) and the expression and secretion of C-X-C motif ligand CXCL2 and CXCL8. Visfatin-treated human EC exhibits increased monocyte adhesion accompanied by the upregulation of adhesions molecules ICAM-1 and VCAM-1 via the NF-κB pathway [107]. In mouse vascular EC, visfatin dramatically downregulated the expression of tight junction-associated proteins including zonula occludens 1&2 (ZO-1&2) and occludin as well as the adherent junction protein VE-cadherin [108]. Thus, visfatin exerts pro-inflammatory effects in leukocytes and microglia, and perturbations in circulating visfatin levels might significantly impact endothelial function.

There is only one study evaluating changes of visfatin in MS, which observed increased plasma visfatin levels in MS patients, despite decreased fat mass. Visfatin levels were highest in RRMS compared to PPMS and SPMS patients and correlated positively to TNF-α and IL-1β [64]. In contrast, newly diagnosed patients showed no differences in visfatin levels [29].

### 3.6. Apelin

Apelin, acting via the apelin receptor (APLNR), is considered anti-inflammatory and plays a significant role in the development and function of macrophages [109]. Apelin downregulates the expression of TNF-α and IL-6 and reduces phagocytic activity in rat peritoneal macrophages in vitro [110]. In mouse macrophages, apelin attenuates the expression of TNF-α, MCP-1, IL-6, and macrophage inflammatory protein (MIP)1α, but it does not modulate the expression of endothelial cell adhesion molecules such as VCAM-1, ICAM-1, and E-selectin as seen for visfatin [111]. Apelin-13 also reduces the secretion of IL-1β and TNF-α and inhibits lipid accumulation in THP-1 macrophages [112]. Furthermore, pre-treatment with [Pyr1]-Apelin-13 significantly diminishes LPS-induced mRNA expression of IL-1β and protein secretion of IL-6 in J774.1 macrophages [113]. Similarly, apelin was shown to attenuate acute LPS-induced lung injury, cytokine expression, and ROS formation, via the inhibition of NF-κB translocation and NLRP3 inflammasome pathways [114]. In contrast, it is shown that apelin upregulates TNF-α, IL-1β, MCP-1, and MIP-1α in BV2 microglia, which was blocked by an inhibitor of the MAPK/ERK signaling pathway [115].

APLNR is expressed on human T and B cells, and the stimulation of T cells with apelin reduces the expression and production of choline acetyltransferase and subsequent acetylcholine release, inhibiting the immune response [116]. In the N9 microglia cell line, apelin-13 slightly attenuates the LPS-induced levels of iNOS and IL-6 and upregulates the level of anti-inflammatory factor arginase 1 (arg-1) and IL-10 and decreased the expression of pSTAT3 [117].

While evidence is emerging that apelin could be a potent anti-inflammatory and neuroprotective compound, only two studies determined blood apelin levels of MS patients. One (incompletely described) study showed that serum levels were increased in RRMS patients compared to controls but showed no correlations with EDSS and disease duration. Unfortunately, no BMI data were reported [118]. The second study observed a decrease in blood apelin levels in females in a very early stage of RRMS (on average less than 1 year after onset), which correlated positively with both EDSS and number of relapses [46].

### 3.7. Confounding Factors

Multiple studies conducted in the past decades investigated the roles and possible treatment opportunities for adipokines in MS. Adiponectin might be a suited biomarker for disease progression, and leptin levels generally tend to increase during or preceding a relapse and correlate positively to pro-inflammatory markers. Studies on resistin, chemerin, visfatin, and apelin are sparse, which makes it hard to draw firm conclusions. Steps forward are hampered by the variation in study outcomes, which is partly explained by the large heterogeneity (including BMI variations) within the MS population and low inclusion numbers. Although it is well-established that females have a higher percentage of WAT, and circulating adipokine levels differ between females and males [119], studies do not always control for gender. Leptin and chemerin levels are increased in females compared to males, which makes it tempting to speculate that these altered levels of pro-inflammatory adipokines link to the increased incidence of the RRMS form of MS in females. Furthermore, gender-specific associations of gene polymorphisms with disease susceptibility have been found in MS patients [44]. In addition, cellular responses to adipokines might differ between gender, since more and more cell types are being discovered of which their function and phenotype is dictated by gender [120,121,122].

Comparisons between studies are often also confounded by age, which has a significant impact on adipokine levels. Nearly all adipokine levels are elevated in older people compared with younger individuals having the same body fat percentage [123]. Another important potential confounder in numerous studies is a lack of controlling for treatment status, which can significantly affect the levels of adipokines, since many treatments are based on immunosuppression [124]. Finally, the interaction between different adipokines is not taken into account. As an example, leptin and adiponectin show to correlate negatively, and the leptin/adiponectin ratio has been proven to be a functional biomarker for adipose tissue inflammation [125], and an increase in this ratio has been related with reduced atherosclerosis risk as well as with a decreased risk of some types of cancer [126].

In conclusion, the adipokines described above have widespread effects on the immune system and significantly affect inflammatory events. Several pro-inflammatory cytokines stimulate the production of pro-inflammatory adipokines while decreasing anti-inflammatory adipokine levels (Table 1). Changes in adipokines in MS patients are evident and suggest that interfering with their signaling pathways could be a novel, interesting treatment strategy for MS.

AdipoR, adiponectin receptor; APLNR, apelin receptor; BBB, blood–brain barrier; BMI, body mass index; CAP1, adenylyl cyclase-associated protein 1; CCL2, C-C Motif Chemokine Ligand 2; CIS, clinically isolated syndrome; CMKLR1, Chemerin Receptor 23; CRP, C-reactive protein; CSF, cerebrospinal fluid; EDSS, extended disability status scale; GPR1, G protein-coupled receptor; IFN, interferon; IL-1β, interleukin 1 β; InsR, insulin receptor; LepR/OB-R, leptin receptor; PBMC, peripheral blood mononuclear cells; PPAR, peroxisome proliferator-activated receptors; PPMS, primary progressive multiple sclerosis; proliferator-activated receptor; RRMS, relapse-remitting multiple sclerosis; SPMS, secondary progressive multiple sclerosis; TLR4, Toll-like receptor 4; TNF-α, tumor necrosis factor α. [127] Range of plasma levels, meta-analysis of a total of 4852 healthy individuals aged 18–59. [128] Mean ± SEM plasma levels, measured by ELISA in 76 healthy individuals, mean age 53.7. [129] Mean ± SD plasma levels, measured by ELISA in 60 healthy individuals mean age 50.7. [130] Mean ± SD plasma levels, measured by radioimmune assay in 25 healthy individuals mean age 57.

## 4. Adipokines as Therapeutic Target for MS—Evidence from Pre-Clinical Studies

Experimental Autoimmune Encephalomyelitis (EAE) has been widely accepted as an animal model for MS, since it resembles many pathological aspects of the disease. The model is typically induced by either active immunization with myelin-derived proteins or peptides, such as myelin basic protein or myelin oligodendrocyte glycoprotein (MOG), in adjuvant or by passive transfer of activated myelin-specific CD4+ T lymphocytes. Depending on the species and strain, animals develop a relapse-remitting or chronic disease course. EAE has been used to test the therapeutic effects of several adipokines [131]. We will here describe proof of concept in vivo and in vitro studies that assessed the effects of adipokines in this animal model.

### 4.1. Adiponectin

Although adiponectin as a treatment has not been tested so far in animal models for MS, adiponectin knock-out (KO) mice showed exacerbated behavioral disability and increased inflammatory cell infiltration and demyelination in the CNS. In vitro, lymphoid cells derived from adiponectin KO mice show higher pro-inflammatory cytokine (IFN-γ, IL-17, TNF-α, IL-6) production in response to MOG compared to cells derived from wild-type mice [132]. Another study found that Th1 and Th17 cell-related cytokines were significantly increased in the lymph nodes of adiponectin-deficient EAE mice, while Th2 cytokines were not affected. In line, adiponectin treatment inhibits Th1 and Th17 differentiation in vitro [133].

The potency of adiponectin in anti-inflammation as well as neuroprotection is also shown in studies of other acute and chronic neurological diseases such as intracerebral hemorrhage [134,135] and Alzheimer disease (AD) [136].

### 4.2. Leptin

The pro-inflammatory role of leptin is evidenced by the findings that leptin is required for EAE induction (i.e., leptin-deficient ob/ob mice are not susceptible for EAE [137]) and intraperitoneal (i.p.) injection of leptin worsened EAE [138], while leptin neutralization improves behavioral disease symptoms [66]. Furthermore, endothelial-specific LepR KO mice show a reduced leukocyte infiltration and inflammation in the CNS and preserved tight junction protein expression during EAE, pointing toward a detrimental role of leptin signaling at the BBB [139]. In contrast, whole-body LepR KO mice (db/db), which display high circulating leptin levels but no intracellular leptin signaling, show BBB dysfunction already at a young age [140].

In two studies using an ischemia/reperfusion animal model, both rodents and macaques, intracisternal or i.p. leptin treatment reduced infarct volume by sustaining BBB permeability, reduction of brain infiltration neutrophils, and neutrophil adherence to vascular ECs [141,142]. Additionally, leptin signaling in astrocytes also seems to be beneficial in EAE animals; an astrocyte-specific lepR KO model worsened disease symptoms and increased T cell infiltration in the CNS [143]. Next to the effects on the BBB, leptin also stimulates oligodendrocyte precursor cell proliferation in vivo [144] and exerts neuroprotective effects [145]. These discrepancies need further investigation but suggest that peripherally, leptin has detrimental effects probably via its pro-inflammatory effects on circulating immune cells, while in the CNS, leptin exerts protective effects on brain endothelial cells as well as astrocytes.

### 4.3. Resistin

The role of resistin in MS pathology in animal models thereof has not been studied so far. However, in wild-type mice, intracerebroventricular (ICV) injection of resistin decreased the amount of GFAP-positive cells, and in vitro, resistin inhibited astrocyte differentiation from stem cells, which was mediated through TLR4. Resistin also significantly decreased BBB integrity and increased permeability in an in vitro co-culture of human brain endothelial cells and astrocytes [146]. In SH-SY5Y human neuroblastoma cells, resistin decreased serum starvation-induced autophagy and increased TNF-α and IL-6 mRNA. The silencing of TLR4 completely abolished these effects. In vivo, ICV resistin recapitulated these results in wild-type but not in TLR4-KO mice [147].

The use of resistin inhibitors has been proposed as a therapeutic approach for MS [148]; however, to our knowledge, no inhibitors have been developed yet, and the absence of a specific resistin receptor complicates the specific targeting of this adipokine.

### 4.4. Chemerin

EAE mice show the expression of chemokine-like receptor (CMKLR1) in microglia, CNS-infiltrating myeloid cells, and dendritic cells isolated from the spinal cord, and chemerin mRNA was upregulated in the spinal cord of these animals. Additionally, CMKLR1-KO mice develop less severe behavioral and histologic EAE signs than their wild-type counterparts. CMKLR1-KO lymphocytes produce pro-inflammatory cytokines in vitro, yet MOG (35–55)-reactive CMKLR1-KO lymphocytes were deficient in their ability to induce EAE by adoptive transfer to wild-type or KO recipients. Moreover, CMKLR1-KO recipients failed to fully support EAE induction by transferred MOG-reactive wild-type lymphocytes [149]. These results further support the involvement and targeted therapeutic possibility of CMKLR1 activation in both the induction and resolution phases of MS.

### 4.5. Visfatin

Targeting visfatin (or NAMPT) as anti-inflammatory therapy has been shown to be effective in clinical trials on NAMPT inhibitors (such as FK866) in melanoma and lymphomas [150]. Pre-clinical studies show that FK866 protects against traumatic brain injury in rats; i.p. treatment limited neural apoptosis, inflammation, and gliosis in the CNS via MAPK/p38 and NF-κB suppression, already after one day of treatment [151]. In mice, spinal cord injury increased NAMPT levels and i.p. treatment with FK866 at 1 h and 6 h after injury, rescued motor function, preserved perilesional gray and white matter, restored anti-apoptotic and neurotrophic factors, prevented the activation of neutrophils, microglia, and astrocytes, and inhibited the elevation of NAMPT, TNF-α, IL-1β, and NF-κB activity [152]. Similar protective findings have been observed in the brain of ischemic rats that received ICV treatment with FK866 for 10 days. The observed NAMPT increase was localized in microglia, and in vitro, FK866 inhibits NF-κB activation in BV2 microglia, suggesting that its protective actions are mediated via microglia [153]. Additionally, NAMPT inhibition also reduces BBB disruption in ischemic rats, as shown by decreased extravascular leakage of the 1 kDa fluorescently labeled cadaverine and larger-sized endogenous IgGs into brain parenchyma after treatment [154].

Only one study investigated the effects of NAMPT inhibition in EAE mice [155]. Treatment with FK866 after symptom onset significantly reduced the disability and demyelination of the spinal cord. In vitro, FK866 prevented T cell proliferation and selectively ablated activated T cells, probably through NAMPT-induced inhibition of NAD+ and subsequent ATP depletion.

In line with these intracellular effects of NAMPT inhibition, neuron-specific deletion of intracellular NAMPT causes neuronal degeneration and led to muscle atrophy, paralysis, and eventually death in mice. Metabolomics of the cortical tissue of these mice revealed that NAMPT deletion activates apoptotic, inflammatory, and immune-responsive pathways and inhibits pathways involved in neuronal/synaptic function [156]. This supports the differential role of intracellular NAMPT as a key enzyme in cell metabolism and extracellular NAMPT as a signaling, cytokine-like molecule to activate inflammatory pathways.

### 4.6. Apelin

Apelin has not yet been studied in animal models of MS; however, it has been shown to suppress neuroinflammation and cognitive decline in an AD rat model [157,158]. These effects were mediated by the brain-derived neurotrophic factor (BDNF)/tropomyosin receptor kinase-B (Trk-B) pathway in the hippocampus [157], which is in line with the activation of this pathway and improved cognitive decline after apelin treatment in rats with cisplatin-induced cognitive dysfunction [159]. The protective effects of apelin have also been shown in brain ischemia. ICV or intranasal delivery of apelin-13 reduced infarct volume and cognitive deficits through the inhibition of microglia recruitment, suppression of inflammatory and cell-adhesion molecules, reduction of ROS, and prevention of apoptosis [160,161,162]. Interestingly, it was also shown that these neuroprotective effects were mediated via the activation of nuclear factor erythroid 2-related factor 2 (Nrf2) [162], which is a direct target of dimethyl fumarate (tecfidera), which is a first-line treatment for MS that reduces relapse rates, the number of lesions, and slows down disease progression [163].

## 5. Conclusions

Numerous studies indicate a possible correlation with different forms of MS, progression, and disability status and levels of adipokines. In MS, adipokines may play a role as potential biomarkers both for the early detection of the disease and for the monitoring of disease activity during disease-modifying therapies. However, steps forward are hampered by the variation in study outcomes, which is partly explained by the large heterogeneity within the MS population. Future (pre-clinical) studies should focus on providing a comprehensive overview of the levels of well-studied adipokines and interactions thereof as well as the newly discovered chemerin, visfatin, and apelin in large and well-defined patient cohorts, which are controlled for BMI, gender, age, and treatment. Furthermore, few studies look at correlations between adipokine levels and described (bio)markers for neuroinflammation and neurodegeneration, such as neurofilament light, as well as clinical outcomes other than disability, such as brain atrophy [164]. Looking at correlations between adipokines and these factors rather than just nominal differences between two groups may reveal novel leads for the use of adipokines as biomarkers for disease. Additionally, the inclusion of longitudinal cohorts will provide answers to whether or not adipokines can be used as predictors for progression and disease severity.

The potent effects of interfering with adipokine signaling and the underlying mechanisms should be investigated further in the context of pathophysiological processes as seen in MS. Findings point to an important role of leptin and visfatin in the exacerbation of MS-related pathology and of adiponectin and apelin in the amelioration of neuroinflammation and neurodegeneration. Investigating the effects of adiponectin and apelin as combination therapy, and/or together with blocking leptin and visfatin signaling, could provide a basis for future clinical studies in MS patients. Additionally, activation of the Nrf2 pathway by apelin could boost the positive effects of the first-line treatment dimethyl fumarate. Another way to target different adipokines concomitantly is by the use of PPAR agonists, as they have been demonstrated to increase levels of anti-inflammatory adipokines and decrease levels of pro-inflammatory adipokines [43,101]. Several PPAR agonists have been shown to ameliorate EAE symptoms by targeting both immune cells as well as neurodegenerative processes [165]. Whether this is partly driven by altering adipokine profiles remains to be investigated. Lastly, there is emerging evidence from human and animal studies that show that diet interventions could positively affect disease course. A high-fat diet that leads to metabolic syndrome and is accompanied by a pro-inflammatory adipokine profile worsens EAE in mice, while caloric restriction attenuates clinical symptoms and reduces immune cell infiltration and demyelination in the CNS, which is accompanied by increased adiponectin and corticosterone levels and a decrease in leptin and IL-6 [166]. In line, intermittent fasting reduced clinical signs accompanied with reduced T cell infiltration and demyelination in EAE mice [167] and also attenuates demyelination in the cuprizone mouse model, where toxic demyelination is induced without adding an immune component [168].

Despite the increasing understanding of the role of adipokines in MS, further investigation should seek to answer the following questions: (1) What are the exact changes in circulating adipokine levels in a well-controlled patient cohort, and how do altered adipokine profiles correlate with known biomarkers of inflammation, neurodegeneration, disability, and progression? (2) Via what mechanisms do adipokines affect pathological processes, including neuroinflammation and neurodegeneration in MS? (3) How can we target adipokine receptors to halt disease? Taking steps forward in answering these key questions will facilitate the development of new therapies, potentially targeting adipokines, for the treatment of MS and its inflammatory and neurodegenerative complications.

## Figures and Tables

**Figure 1 ijms-22-10845-f001:**
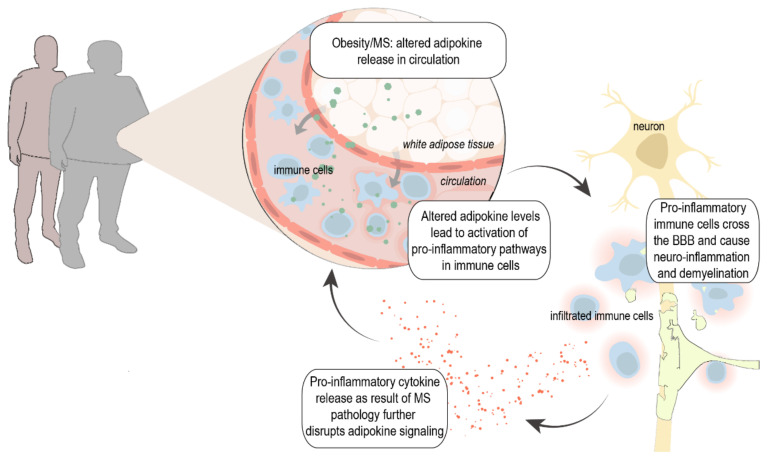
Proposed hypothesis: Obesity and MS can lead to altered adipokine release from white adipose tissue (WAT) into circulation. This results in the activation of inflammatory pathways and increased infiltration of immune cells in the CNS. On the other hand, increased levels of pro-inflammatory cytokines in MS such as tumor necrosis factor α (TNF-α), interleukin 6 (IL-6), and interferon γ (IFN-γ) further stimulate the release of pro-inflammatory adipokines from the WAT and disrupt adipokine signaling, creating a detrimental positive feedback loop. MS, multiple sclerosis; BBB, blood–brain barrier.

**Table 1 ijms-22-10845-t001:** Overview of adipokines, receptors, levels in circulation, role in immunity, and plasma and CSF levels in MS.

Adipokine	Primary Production Site (s)	Target Receptor (s)	Levels in Circulation	Role in Immunity	Changes in Circulating Levels in MS
Adiponectin	Adipocytes	AdipoR1 and 2	2.5–22 µg/mL [127]	∙ Anti-inflammatory effects on T cells and endothelial cells ∙ Pro-inflammatory cytokines decrease adiponectin levels ∙ Anti-inflammatory agents (PPAR agonists) increase adiponectin levels	↑ early-onset RRMS females versus control [46] ↑ during remission [45] ↑ after IFN-β therapy [49] ↑ treatment-naïve patients, correlated with increased disability and progression [50] ↑ **CSF**, predicted worse prognosis and higher EDSS [52,53] ↔ newly diagnosed, treatment-naive patients ↓ after BMI adjustment [29] ↔ between CIS and other MS types [85] ↓ female MS versus control [48] ↓ RRMS females in relapse phase versus control [44] ↓ in patients with optic neuritis as first clinical episode [51]
Leptin	Adipocytes	LepR/OB-R	5–50 ng/mL [127]	∙ Pro-inflammatory effects on T-, B cells and macrophages ∙ Preserves BBB function ∙ Protective in astrocytes	*Levels:* ↑ in MS patients versus control [62,64,70] ↑ in MS patients in relapse phase versus control [44,47,66,71] ↑ CSF in relapse RRMS versus control [66] ↑ in MS patients in remission phase versus control [51,72] ↔ between CIS and other MS types [85] ↔ in MS patients in relapse phase versus control [67] ↔ in MS patients in remission phase versus control [53,63,65,67,68,69,73] ↔ **CSF** in MS patients in remission phase versus control [53] ↓ in MS patients in relapse phase versus control [63,72] *Correlations:* ↑ plasma with EDSS in RRMS [75] ↑ plasma with EDSS in SPMS/PPMS [62,76] ↑ plasma with less T-reg cells [47,64,66] ↑ plasma correlated with TNF-α, IL-1β, CRP [64] ↔ no correlation with EDSS scores in RRMS patients [49,69,73,74]
Resistin	PBMC, adipocytes	CAP1, TLR4	38.78 ± 7.9 ng/mL [128]	∙ Pro-inflammatory in human PBMC and macrophages ∙ Pro-inflammatory effects on endothelial cells ∙ Pro-inflammatory cytokines increase resistin production in PBMC	↑ plasma, correlated with pro-inflammatory cytokines and EDSS ↔ between CIS and other MS types [85]
Chemerin	Adipocytes, hepatocytes	CMKLR1, GPR1, CCRL2	62.1 ± 19.2 ng/mL [129]	∙ Inflammation initiation; recruitment of immune cells to site of inflammation ∙ Resolution; polarization of macrophages toward anti-inflammatory phenotype	↔ between MS and control [97,98]
Visfatin	Adipocytes, macrophages, endothelial cells	InsR, TLR4	11.0 ± 2.0 ng/mL [130]	∙ Pro-inflammatory in macrophages, microglia, and endothelial cells ∙ Pro-inflammatory cytokines increase visfatin ∙ Anti-inflammatory PPAR-γ agonist rosiglitazone reduced visfatin expression by adipocytes [101]	↑ RRMS compared to PPMS and SPMS patients, positive correlation with TNF-α and IL-1β [64] ↑ newly diagnosed, treatment-naive patients <-> after BMI adjustment [29]
Apelin	Adipocytes	APLNR	205 ± 108 pg/mL [130]	∙ Anti-inflammatory in macrophages, no effect on adhesion molecules ∙ Protects against hypoxia-related changes in macrophages ∙ Attenuates T cell activation ∙ Pro-inflammatory in microglia	↑ RRMS patients compared to controls, no correlations with EDSS and disease duration [118] ↓ RRMS females less than 1 year after onset, levels correlated positively with both EDSS and number of relapses [46]

## Data Availability

Not applicable.

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
