# Peer review of "Adipokines as Immune Cell Modulators in Multiple Sclerosis"

_ijms, 2021, doi:10.3390/ijms221910845_

Round 1

Reviewer 1 Report

Dear Authors,

The role of adipokines in multiple sclerosis is indeed an attractive area of research. Your review article “Adipokines as immune cell modulators in multiple sclerosis” is interesting and may have impact on this area of research. The manuscript is well written and understandable.

I have only minor suggestions for improving the manuscript:

The sections “Diagnosis and pathophysiology” and “Disease modifying treatments” do not fully correspond to the topic of the review article and contain generally known information. These sections can be shortened or removed from the text. But if the authors think otherwise, then you can leave everything as it is.

Line 3 – “and” – please replace with “,”.

Line 31– “neuro-inflammation” – correct please.

Line 47 - Typically, the figure follows the paragraph immediately after the first mention.

Line 55 – “Diagnosis and pathophysiology” – All subsections must be numbered. Correct throughout the text, please.

Line 85 – “IL6” – Typically, this abbreviation refers to the gene encoding interleukin 6. The protein interleukin 6 is commonly referred to as “IL-6”. Other interleukins are labeled similarly. Please replace throughout the text.

Line 95 – “blood–brain barrier (BBB)” – This abbreviation has already been defined. Please see line 76.

Line 116 – “food and drug administration” - Please replace with “Food and Drug Administration”.

Lines 115-119 – Please rephrase the sentence.

Line 176 – “;” – Please replace with “:”.

Line 208 – “fatty acid burning” – It would be more correct “fatty acid oxidation”.

Line 259 – “NFκB” – define an abbreviation please. Besides, it would be more correct “NF-κB”. Please replace throughout the text.

Line 283 – “nuclear factor kappa-light-chain-enhancer of activated B-cells” – The abbreviation “NF-κB” must have been defined earlier (see line 259).

Lines 457-460 – remove italics.

Line 491 – “in to” – correct please.

Line 590 – “lymphoma’s” – correct please.

The bibliography style does not match the requirements of the journal. Please correct as required.

Good luck with your further research.

Best regards

Author Response

Response to Reviewer 1 comments

We thank the reviewer for critically reviewing our manuscript and providing input on the article.

Point 1: The sections “Diagnosis and pathophysiology” and “Disease modifying treatments” do not fully correspond to the topic of the review article and contain generally known information. These sections can be shortened or removed from the text. But if the authors think otherwise, then you can leave everything as it is.
We feel that a proper introduction of MS pathology and disease modifying treatments are needed for the flow of the review. Nevertheless we appreciate this input and have shortened these sections to draw more attention to the adipokines.

Minor points:

  • Line 3 – “and” – please replace with “,”.
  • Line 31– “neuro-inflammation” – correct please.
  • Line 47 - Typically, the figure follows the paragraph immediately after the first mention.
  • Line 55 – “Diagnosis and pathophysiology” – All subsections must be numbered. Correct throughout the text, please.
  • Line 85 – “IL6” – Typically, this abbreviation refers to the gene encoding interleukin 6. The protein interleukin 6 is commonly referred to as “IL-6”. Other interleukins are labeled similarly. Please replace throughout the text.
  • Line 95 – “blood–brain barrier (BBB)” – This abbreviation has already been defined. Please see line 76.
  • Line 116 – “food and drug administration” - Please replace with “Food and Drug Administration”.
  • Lines 115-119 – Please rephrase the sentence.
  • Line 176 – “;” – Please replace with “:”.
  • Line 208 – “fatty acid burning” – It would be more correct “fatty acid oxidation”.
  • Line 259 – “NFκB” – define an abbreviation please. Besides, it would be more correct “NF-κB”. Please replace throughout the text.
  • Line 283 – “nuclear factor kappa-light-chain-enhancer of activated B-cells” – The abbreviation “NF-κB” must have been defined earlier (see line 259).
  • Lines 457-460 – remove italics.
  • Line 491 – “in to” – correct please.
  • Line 590 – “lymphoma’s” – correct please.
  • The bibliography style does not match the requirements of the journal. Please correct as required.

We thank the reviewer for thoroughly reading the manuscript and have adjusted all minor points stated above as recommended.

Reviewer 2 Report

  1. The paper is far too long.
  2. Chapter 3 should be shorted and combined with Chapter 4. Then, the content should be reedited. Now, Chapter 3 presents data concerning the relationship between adipokines and the immunological system, and Chapter 4 relates to changes in adipokine profile in a course of multiple sclerosis. It would be more clear to present the above data in the following pattern: general information about one selected adipokine, and then describe detailed data on how it changes in multiple sclerosis. The authors should also thoroughly discuss why these changes happen and what are the consequences of this phenomenon. Instead of mentioning and repeating data presented by others, authors should show data in a related and logical manner
  3. In Chapter 5 there is a paragraph concerning adiponectin in Alzheimer's Disease. The text is without any relationship to multiple sclerosis and should be removed.

Author Response

Response to Reviewer 2 comments

We thank the reviewer for her/his critical view and providing valuable input on our manuscript.

Point 1: The paper is far too long.

In line with point 2 of this reviewer, we have shortened Chapter 3 and 4 and critically looked at the rest of the review, and shortened it where possible (for instance chapters on “Diagnosis and pathophysiology” and “Disease modifying treatments”).

Point 2: Chapter 3 should be shorted and combined with Chapter 4. Then, the content should be reedited. Now, Chapter 3 presents data concerning the relationship between adipokines and the immunological system, and Chapter 4 relates to changes in adipokine profile in a course of multiple sclerosis. It would be more clear to present the above data in the following pattern: general information about one selected adipokine, and then describe detailed data on how it changes in multiple sclerosis. The authors should also thoroughly discuss why these changes happen and what are the consequences of this phenomenon. Instead of mentioning and repeating data presented by others, authors should show data in a related and logical manner.

We agree with the reviewer that chapter 3 can be shorter and adapted this as such. We also combined and shortened chapter 3 and 4, where we now discuss ”Adipokines, inflammation and MS” for every individual adipokines as the reviewer suggested (first general information about one selected adipokine, and then describe detailed data on how it changes in multiple sclerosis). Why adipokine levels exactly change in MS is unknown, but we discuss possible explanations with the reciprocal interaction between pro-inflammatory cytokines and adipokines as an important factor.

Point 3: In Chapter 5 there is a paragraph concerning adiponectin in Alzheimer's Disease. The text is without any relationship to multiple sclerosis and should be removed.

We have deleted this paragraph from the text. However, since this paper (and the paper on ICH) could be of interest for MS due to the pathways investigated, we decided to just briefly mention it.